# AutoEncoder-Based Anomaly Detection for CMS Data Quality Monitoring

## Abstract

The monitoring of data quality in high-energy physics experiments is essential both during data acquisition and in offline analyses to ensure the reliability of datasets. The Compact Muon Solenoid (CMS) experiment at the Large Hadron Collider (LHC) has recently implemented Data Quality Monitoring (DQM) at the granularity of individual "luminosity sections" (LSs), each representing about 23 seconds of data taking. This paper presents a novel application of AutoEncoders for anomaly detection in DQM, specifically targeting quantities associated with jets and missing transverse energy (MET). The developed method allows for the detection of anomalies at the LS level, which might be missed when examining integrated quantities. By automating the identification of anomalies, this approach enhances the efficiency and precision of the DQM process, ultimately improving the quality of the datasets used for analysis.

## 1 Introduction

The Compact Muon Solenoid (CMS) [CMS Collaboration, 2008] is a general-purpose detector at the Large Hadron Collider (LHC) at CERN. CMS is designed to study high-energy proton-proton collisions to better understand the fundamental forces and particles that make up the Universe. The CMS apparatus is composed of a complex system of sub-detectors to detect the particles produced in a proton or ion collision. The only particles that CMS can not directly detect are neutrinos, because of their very weak interaction with matter. To indirectly observe neutrinos, a kinematics observable called missing transverse energy (MET) is usually employed. MET is defined as:

$$\text{MET} = \left| - \sum_i \vec{p}_{T,i} \right|, \tag{1}$$

where $\vec{p}_{T,i}$ is the transverse momentum of the $i$-th reconstructed particle of the final state.

Since the transverse momentum of the initial state is null, according to the law of conservation of momentum and energy, MET is expected to vanish if all products of a collision were detected. However, because neutrinos and other weakly

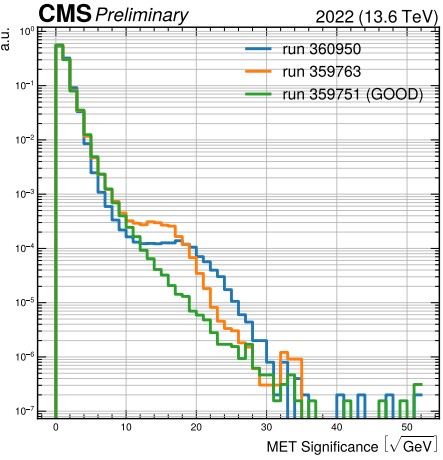

Figure 1: Histograms of a Monitor Element (MET Significance) for three different runs, one flagged *GOOD* and two presenting an anomaly, therefore flagged *BAD*.

interacting particles can escape the detector without being directly detected, their presence result in a non vanishing missing transverse energy value.

Particles that have a colour charge (like quarks and gluons) can not be directly observed as well. This is because a fundamental principle called colour confinement, according to which colour charged particles can not be isolated and they always combine in ways that ensure their overall colour charge is colour neutral. In order to obey colour confinement, quarks and gluons produced in strong interaction processes create other coloured particles to form hadrons clustered in *jets*, i.e. collimated groups of colourless objects [Ali and Kramer, 2011].

LHC is a proton-proton collider. its operation consists of several phases, which can be broken down in three main stages: the filling of the machine with proton beams (which takes minutes); the subsequent collision phase, in which the beams are brought into collision, which can last several hours, typically until the proton population in the beams has fallen below a predefined threshold; the beam dump, in which the remaining beams are dumped and the machine is cycled

again. These three stages are collectively call in jargon a *fill*. CMS takes data during the collision phase of a fill and this data is gathered in "luminosity sections", lumisections in short (LSs), that are sub-sections corresponding to around 23 seconds of data taking during which the instantaneous *luminosity* (a quantity related to the collision rate) is almost constant [CMS Collaboration, 2008]. LSs are grouped in *runs*, of thousands of LSs.

Being CMS composed of various subsystems, each serving a specific purpose in particle detection and measurement, issues in the different sub-detectors can arise due to various factors, such as radiation damage, electronic noise, aging of components and temporary malfunctions (such as tripping of individual components). The monitoring of data quality is therefore crucial both online, during the data taking, to promptly spot issues and act on them, and offline, to provide analysts with datasets that are cleaned against the occasional failures that may have crept in. Data Certification (DC) is the final step of quality checks performed by Data Quality Monitoring (DQM) on recorded collision events. For each run, experts monitor several reconstructed distributions called Monitor Elements (MEs) to spot issues and anomalies in the data. For quantities pertaining to hadronic jets and MET, an issue in a few LSs could cause the entire run to be flagged as problematic (*BAD*) and thus removed from the pool of good-for-analysis data (*GOOD*).

Figure 1 shows the integrated (over the whole run) histogram illustrating a specific ME (MET Significance) for three distinct runs— one categorised as *GOOD* and the other two as *BAD*.

MET Significance is defined as:

$$\text{METSig} \equiv \frac{\text{MET}}{\sqrt{\text{SumET}}} = \frac{MET}{\sqrt{\sum_i |\vec{p}_{T,i}|}} \ . \tag{2}$$

This paper introduces a novel application of AutoEncoders (AEs) for anomaly detection within the CMS DQM framework. By exploiting unsupervised machine learning techniques, we aim to automate the identification of anomalous LSs. This approach enhances the efficiency and precision of the DQM process, allowing for the isolation and removal of problematic LSs, thereby improving the overall quality of datasets available for analysis. Our method demonstrates significant improvements in detecting subtle anomalies and ensures that data previously flagged as problematic can be refined and utilised effectively, ultimately contributing to more accurate and reliable physics analyses.

## 2 Methods

CMS has recently extended the possibility of accumulating quantities monitored for data quality purposes per-LS to Jet and Missing Energy (JME) MEs. This capability allows for a higher granularity detection of anomalies, potentially enabling the saving of higher amounts of data from runs presenting only a limited set of anomalous LSs. Given the high number (order of thousands) of LSs to be analysed for each run, an automated approach for DC is required.

Machine Learning (ML), particularly Neural Networks (NN) [Goodfellow, 2016], can be implemented to this end.

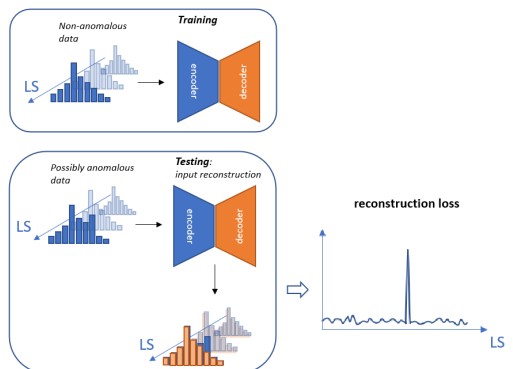

Figure 2: Scheme of training and testing steps for the models

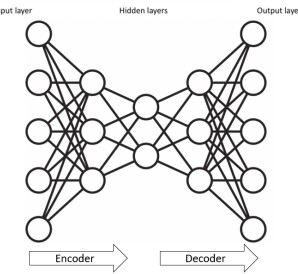

Figure 3: Structure of the dense Under-complete AE (the number of nodes is just indicative)

Therefore, to attack the problem, we employed unsupervised ML models based on AutoEncoders (AE) [Hinton and Salakhutdinov, 2006].

### 2.1 Input data and preprocessing

Given a specific ME, the input features to our models consist of bins of the corresponding histogram, with each LS being a single time sample. Thus, data is structured in the shape $(\#bins, \#LS)$.

Before feeding the models with training (and testing) data we made a rescaling in the $[0, 1]$ interval. This is a common practice for this kind of models. Different rescalings are possible, but one that we found very effective is the following bin by bin rescaling:

$$\hat{x}_{\text{train}} = \frac{x_{\text{train}} - \min(x_{\text{train}})}{\max(x_{\text{train}}) - \min(x_{\text{train}})}, \tag{3}$$

where the maximum and minimum are computed along the time direction.

### 2.2 Models

Two types of AEs were developed: a dense Under-complete AE and a Long Short-Term Memory (LSTM) Under-complete AE.

The first model that was optimised is a dense Under-complete AE [Hinton and Salakhutdinov, 2006] built using dense layers with three hidden layers in total, see Figure 3. The second model is the more complex LSTM Under-complete AE [Wei *et al.*, 2023] schematised in Figure 4. This

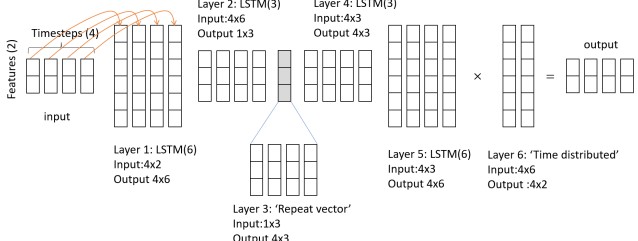

Figure 4: Structure of the LSTM Under-complete AE (the number of nodes is just indicative)

model is designed to handle sequential data, making it suitable for the time-series nature of DQM metrics. The structure is analogous to the dense Under-complete AE, with layers showing again a decrease followed by an increase of the number of nodes but with the complication that each node is an LSTM node, i.e. a Long Short-Term Memory recurrent neural network (RNN). Due to the inherent recurrent nature of LSTM, each node takes as input not a single time sample, but a certain window of them. Thus, the output of each layer is duplicated to enter each of the copies of every node of the following layer. For the latent layer, a `RepeatVector` layer is used to bring copies of the layer to the following decoding layer.

## 2.3 Training and testing

Both the models were trained on non-anomalous data from *GOOD* runs: histograms of specific MEs are fed to the model with per-LS granularity to allow the AE to learn a normal, non-anomalous behaviour of that specific ME, see Figure 2. The training is performed via the minimisation of the reconstruction loss, a measure of the distance between the input and output of the AE. In this case, the reconstruction loss is the mean squared error (MSE):

$$\text{MSE} = \frac{1}{n} \sum_{i=1}^{n} (y_i - \hat{y}_i)^2, \qquad (4)$$

where $y$ and $\hat{y}$ are respectively the input and the output of the AE, and $n$ is the bin number.

Possibly anomalous runs under investigation are tested by examining again the reconstruction loss: peaks in this function indicate LSs containing histograms that deviate from the learned behaviour.

Optimised models (one for each ME) are paired with a threshold value `thr` for the reconstruction loss that has been tuned on a set of known anomalous runs. If the reconstruction loss exceeds this threshold during testing, it is considered anomalous, and the corresponding LSs are removed.

## 3 Results

The models are tested in this example on a run (360950) that was flagged *BAD* by JME due to the presence of an anomaly visible in histograms of many different MEs, see e.g., Figure 1. By analysing the per-LS MET Significance for the run via the dense Under-complete AE, a peak is observed in the

reconstruction loss corresponding to a specific LS (Figure 5). The threshold for this model, $\text{thr}_{\text{dense}} = 0.1$, is passed.

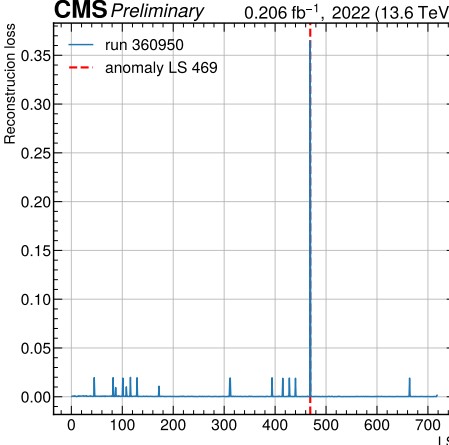

Figure 5: Reconstruction loss by the dense Under-complete model for an anomalous run showing a high peak corresponding to LS 469

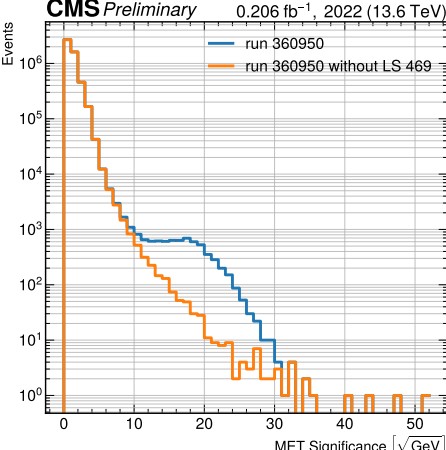

Figure 6: Histogram of an anomalous run before and after the removal of the identified anomalous LS

Once the anomalous LS is identified, it is removed from the run. The resulting histograms for the *BAD* run show how the cause of the MET Significance anomaly was isolated to a specific LS, as shown in Figure 6. The exclusion of the identified anomalous LS results in the remaining data no longer exhibiting the anomaly.

As a second example, we consider a run presenting an analogous anomaly, Figure 7. When tested with the dense Under-complete model, only a major peak in the reconstruction loss is visible, along with smaller peaks not relevant according to the predefined threshold, Figure 8. When the only relevant LS is removed, the resulting histogram still presents an

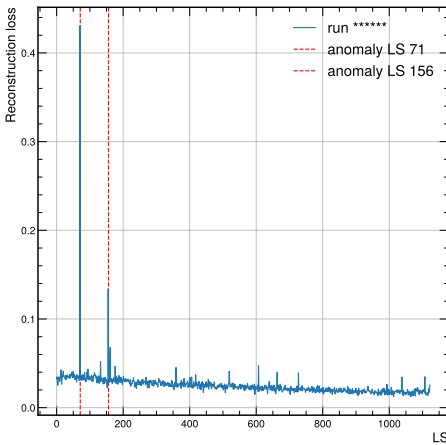

Figure 9: Reconstruction loss by the LSTM Under-complete model for an anomalous run showing a high peak (LS 71) and a second less pronounced peak (LS 156). Both are above our fixed threshold for anomalies

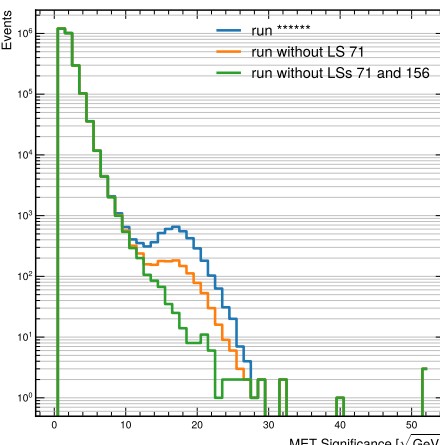

Figure 7: Histogram of an anomalous run before and after the removal of the identified anomalous LSs. The orange histogram represents the result after removing the LS identified by the dense Under-complete model, while the green one shows the result after removing both LSs identified by the LSTM Under-complete model

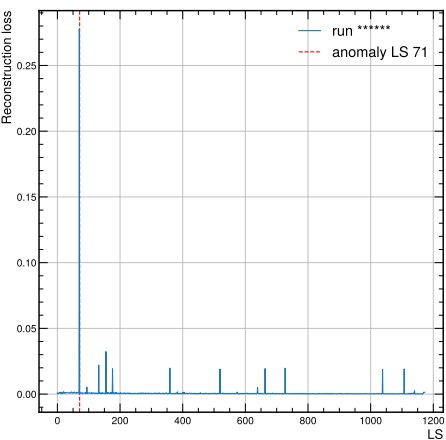

Figure 8: Reconstruction loss by the dense Under-complete model for an anomalous run showing a high peak corresponding to LS 71 above our fixed threshold for anomalies

anomalous shape, Figure 7 . As changing the threshold allows for the removal of the whole anomaly, we decided to test the more complex LSTM Under-complete AE on the run. The resulting reconstruction loss shows a more pronounced peak for a second LS, acceptable according to the threshold for the model, $\mathtt{thr}_{LSTM} = 0.1$, Figure 9 .

The removal of both the major peaks results in the complete cleaning of the anomaly, Figure 7 . When inspecting the two identified LSs, it is apparent that both anomalies were affecting the same set of bins in the histograms, with the second one being less pronounced: this results in a suppression of the magnitude of the rescaled bins after (3), making the anomaly far less visible to the dense Under-complete model.

## 4 Conclusions

An AutoEncoder-based anomaly detection tool has been successfully developed and tested for DQM in the CMS experiment. This tool, capable of detecting anomalies at the per-LS granularity, significantly improves the data certification process by isolating problematic LSs within runs flagged as *BAD*. While some anomalies could be detected by simple comparisons with average values, the models presented, and in particular the LSTM AE, prove versatile and robust across different types of anomalies, enhancing the overall data quality.

The removal of the identified anomalous LSs ensures that the remaining data is reliable, and the recovery of data that would otherwise be discarded. This approach not only streamlines the DQM process but also increases the efficiency and accuracy of data used for physics analyses, demonstrating the potential of machine learning techniques in high-energy physics.

This work uses results that are part of a CMS Detector Performance Note (DP-note) [CMS Collaboration, 2023].

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
