# OpenReview forum: "AutoEncoder-Based Anomaly Detection for CMS Data Quality Monitoring"
_ijcai.org/IJCAI/2024/Workshop/AI4Research — AI4Research 2024_

### Official Review · Reviewer_CLMc · 2024-05-31
**The paper investigates the application of autoencoders for anomaly detection in particle physics data and demonstrates their effectiveness using data from the CMS experiment.**

**Rating:** 4
**Confidence:** 4

**Review:**

**Strengths**
1. The paper explores the application of autoencoders for anomaly detection in particle physics anomaly detection, and directly demonstrates their usefulness in the CMS experiment. Therefore, this work is significant to the particle physics community.

**Weakness**
1. The paper lacks comparisons with traditional methods used in the CMS experiment, making it difficult to assess the relative performance and potential advantages of the proposed approach.
2. The paper lacks quantitative metrics to demonstrate the effectiveness and accuracy of the anomaly detection approach. The performance is only showed via two histogram figures (Figure 6, Figure 7).
3. The paper selects two models for evaluation: (1) a dense autoencoder and (2) an LSTM. However, it does not provide a rationale for the exclusive choice of these models. Additionally, the paper lacks an ablation study to analyze the contributions of individual components or model variations.

**Questions**
1. The input data format consists of histogram bins, yet the number of bins appears to be an important hyper-parameter that is not fully discussed. Have you considered conducting ablation studies on the number of bins or combining this approach with a density smoothing technique, such as density estimation?
2. Could you clarify whether you used the same Run (360950) for both training and testing? Additionally, could you comment on the potential of overfitting in this context?

---

### Official Review · Reviewer_XKp7 · 2024-06-02
**Interesting idea but experiment is not solid**

**Rating:** 6
**Confidence:** 2

**Review:**

This paper introduces a novel application of AutoEncoders (AEs) for anomaly detection within the Data Quality Monitoring (DQM) framework of the Compact Muon Solenoid (CMS) experiment at the Large Hadron Collider (LHC). The primary focus is on improving the granularity of anomaly detection for quantities associated with jets and missing transverse energy (MET) at the level of individual luminosity sections (LSs). By automating the identification of anomalies, the proposed method aims to enhance the efficiency and precision of the DQM process, ultimately improving the quality of the datasets used for analysis.

Pros of the Methods
1. The method allows for anomaly detection at the LS level, which provides finer granularity compared to traditional methods that examine integrated quantities.
2. The use of AutoEncoders automates the anomaly detection process, reducing the reliance on manual monitoring by experts and increasing the efficiency of the DQM process.
3. The unsupervised nature of the method eliminates the need for labeled training data, which is particularly beneficial in scenarios where obtaining labeled anomalies is challenging.


Cons of the Methods
1. The implementation of LSTM-based AutoEncoders, while effective, introduces additional complexity in terms of model training and parameter tuning compared to simpler dense AutoEncoders.
2. The method relies on predefined thresholds for the reconstruction loss to identify anomalies. Setting these thresholds appropriately can be challenging and may require extensive tuning.
3. The effectiveness of the models may vary depending on the specific characteristics of the data and the types of anomalies present. There is a potential risk that the models may not generalize well to all types of anomalies or other datasets.

---

### Decision · Program_Chairs · 2024-06-03

Accept